# Optimal recovery of missing values for non-negative matrix factorization: A probabilistic error bound

**Rebecca Chen** [1]   **Lav R. Varshney** [1,2]

## Abstract

Missing values imputation is often evaluated on some similarity measure between actual and imputed data. However, it may be more meaningful to evaluate downstream algorithm performance after imputation than the imputation itself. We describe a straightforward unsupervised imputation algorithm, a minimax approach based on optimal recovery, and derive probabilistic error bounds on downstream non-negative matrix factorization (NMF). We also comment on fair imputation.

## 1. Introduction

The performance of missing values imputation is typically measured by how similar imputed data is to original data, but Tuikkala et al. (2008) claim it is more meaningful to measure the accuracy of downstream analysis (e.g. clustering accuracy) after imputation. Several groups have evaluated the effects of different imputation methods on clustering and classification accuracy (Chiu et al., 2013; de Souto et al., 2015). We (2019) extended this to quantify non-negative matrix factorization (NMF) accuracy after imputation, and introduced a new imputation technique. In this paper, we find a probabilistic bound for NMF error.

NMF is a popular clustering algorithm used by scientists because of its identifiability, or model uniqueness, and its interpretability, with resulting latent factors being intuitive (Li & Ngom, 2013; Qi et al., 2009). NMF has been used for speech and audio separation, community detection, and topic modeling. In biology, NMF of gene count matrices can discover cell groups and lower-dimensional manifolds (latent factors) describing gene count ratios for different cell types. However, due to channel noise, incomplete survey data, or biological limitations, data matrices are usually incomplete and matrix imputation is often needed before further analysis (Li & Ngom, 2013). In particular, "newer MF algorithms that model missing data are essential for [single-cell RNA] data" (Stein-O'Brien et al., 2018).

We took up this challenge using *optimal recovery* (Golomb & Weinberger, 1959; Micchelli & Rivlin, 1976; 1985), an estimation-theoretic approach used for signal and image interpolation (Donoho, 1994; Muresan & Parks, 2004; Shenoy & Parks, 1992). We previously found a tight *worst-case* bound on NMF error following our minimax imputation, and experiments showed competitive performance with more complicated imputation techniques (Chen & Varshney, 2019). In this paper, we find several *probabilistic* error bounds, which would better characterize experimental results and serve as useful benchmarks for algorithms. We made no assumptions on the missingness pattern for the worst-case bound (Little & Rubin, 2002), but we assume samples are missing completely at random (MCAR) for our probabilistic bounds. Finally, we discuss how our minimax approach aligns with certain notions of fairness.

## 2. Related Work

### 2.1. Non-negative matrix factorization (NMF)

When used to perform cluster analysis, NMF outputs latent factors that characterize the clusters. Donoho & Stodden (2004) interpret NMF as the problem of finding cones in the positive orthant which contain clouds of data points. Liu & Tan (2017) show that a rank-one NMF gives a good description of near-separable data and provide an upper bound on the relative reconstruction error. Given certain biological data is often linearly separable on some manifold- or high-dimensional space (Clarke et al., 2008), the bound given by rank-one NMF is valid. We describe this conical model below (Donoho & Stodden, 2004; Fu et al., 2019; Liu & Tan, 2017).

Let $\mathbf{V} \in \mathbb{R}_+^{F \times N}$ be a matrix of $N$ sample points with $F$ non-negative observations. Suppose the columns in $\mathbf{V}$ are

[1]Department of Electrical and Computer Engineering, University of Illinois at Urbana-Champaign, Urbana, Illinois, USA. [2]Salesforce Research, Palo Alto, CA, USA

This work was supported by Air Force STTR Grant FA8650-16-M-1819 and grant number 2018-182794 from the Chan Zuckerberg Initiative DAF, an advised fund of the Silicon Valley Community Foundation.. Correspondence to: Lav R. Varshney <varshney@illinois.edu>.

*Presented at the first Workshop on the Art of Learning with Missing Values (Artemiss) hosted by the $37^{th}$ International Conference on Machine Learning (ICML).* Copyright 2020 by the author(s).

generated from $K$ clusters. There exist $\mathbf{W} \in \mathbb{R}_+^{F \times K}$ and $\mathbf{H} \in \mathbb{R}_+^{K \times N}$ such that $\mathbf{V} = \mathbf{WH}$. This is the NMF of $\mathbf{V}$ (Lee & Seung, 1999).

Suppose the $N$ data points originate from $K$ cones. We define a circular cone $C(u, \alpha)$ by a direction vector $u$ and an angle $\alpha$:

$$C(u, \alpha) := \left\{ x \in \mathbb{R}^F \backslash \{0\} : \frac{x \cdot u}{\|x\|_2} \geq \cos \alpha \right\}. \quad (1)$$

We truncate the circular cones to be in the non-negative orthant $P$. All $x$'s belonging to $C_k$ can be considered as noisy versions of $u_k$, where $k = 1, \ldots, K$. We call the angle between cones $\beta_{ij} := \arccos(u_i \cdot u_j)$. Assume that

$$\min_{i,j \in [K], i \neq j} \beta_{ij} > \max_{i,j \in [K], i \neq j} \{\max\{\alpha_i + 3\alpha_j, 3\alpha_i + \alpha_j\}\}. \quad (2)$$

This is a common assumption used to guarantee clustering performance (Bu et al., 2017; Liu & Tan, 2017; Ng et al., 2001). We can then partition $\mathbf{V}$ into $k$ sets, denoted $\mathbf{V}_k := \{\mathbf{v}_n \in C_k \cap P\}$, and rewrite $\mathbf{V}_k$ as the sum of a rank-one matrix $\mathbf{A}_k$ (parallel to $u_k$) and a perturbation matrix $\mathbf{E}_k$ (orthogonal to $u_k$). For any vector $\mathbf{z} \in \mathbf{V}_k$, $\mathbf{z} = \|\mathbf{z}\|_2 (\cos \beta) \mathbf{u}_k + \mathbf{y}$, where $\|\mathbf{y}\|_2 = \|\mathbf{z}\|_2 (\sin \beta) \leq \|\mathbf{z}\|_2 (\sin \alpha_k)$. This rank-one approximation is used to find error bounds (Liu & Tan, 2017).

## 2.2. Optimal recovery imputation

When there are missing values in clustered data, local imputation approaches, or those based on local patterns (such as cluster membership or closest neighboring points), can be used to impute values. Popular algorithms that utilize local structure include k-nearest neighbors, local least squares, and bicluster Bayesian component analysis (Hastie et al., 1999; Kim et al., 2005; Meng et al., 2014).

Optimal recovery imputation minimizes NMF error under certain geometric assumptions on data and enables error bound derivation (Chen & Varshney, 2019). Suppose we are given an unknown signal $v$ that lies in some signal class $C_k$. The optimal recovery estimate $\hat{v}$ minimizes the maximum error between $\hat{v}$ and all signals in the feasible signal class. Given well-clustered non-negative data $\mathbf{V}$, we impute missing samples in $\mathbf{V}$ so the maximum error is minimized over feasible clusters, regardless of the missingness pattern.

Suppose there are missing values in $\mathbf{V}$. Let $\Omega \in \{0, 1\}^{F \times N}$ be a matrix of observed values indicators with $\Omega_{ij} = 1$ if $v_{ij}$ is observed and 0 otherwise. We define the projection operator of a matrix $\mathbf{Y}$ onto an index set $\Omega$ by

$$[P_\Omega(\mathbf{Y})]_{ij} = \begin{cases} \mathbf{Y}_{ij} & \text{if} \quad \Omega_{ij} = 1 \\ 0 & \text{if} \quad \Omega_{ij} = 0 \end{cases}.$$

We use the subscripted vector $(\cdot)_{fo}$ to denote fully-observed data points (columns), or data points with no missing values,

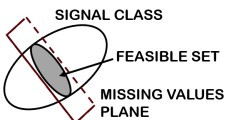

*Figure 1.* Feasible set of estimators.

and we use the subscripted vector $(\cdot)_{po}$ to denote partially-observed data points. We use a subscripted matrix $(\cdot)_{fo}$ or $(\cdot)_{po}$ to denote the set of all fully-observed or partially-observed data columns in the matrix.

We can impute a partially-observed vector $v_{po}$ by seeing where its observed samples intersect with the clusters $C_1, ..., C_k$, as shown in Fig. 1. Let the *missing values plane* be the restriction set over $\mathbb{R}^F$ that satisfies the constraints on the observed values of $v_{po}$. We call this intersection the *feasible set $W$*:

$$W = \bigcup_{k=1}^{K} \{\hat{v}_{po} \in C_k : P_\Omega(\hat{v}_{po}) = P_\Omega(v_{po})\}. \quad (3)$$

For some norm or error function, $\| \cdot \|$, the optimal recovery estimator $\hat{v}_{po}^*$ minimizes the maximum error over the feasible set of estimates:

$$\hat{v}_{po}^* = \arg\min_{\hat{v}_{po} \in C_k} \max_{v \in C_k} \|\hat{v}_{po} - v\|. \quad (4)$$

If $W$ contains estimators belonging to more than one $C_k$, $W$ can be partitioned into disjoint sets $W_k$. Feasible clusters are those for which $W_k$ is not empty.

If the columns in $\mathbf{V}$ come from $K$ circular cones defined as (1), there is a pair of factor matrices $\mathbf{W}^* \in \mathbb{R}_+^{F \times K}, \mathbf{H}^* \in \mathbb{R}_+^{K \times N}$, such that

$$\frac{\|\mathbf{V} - \mathbf{W}^* \mathbf{H}^*\|_F}{\|\mathbf{V}\|_F} \leq \max_{k \in [K]} \{\sin \alpha_k\}. \quad (5)$$

The optimal recovery estimator $\hat{v}_{po}^*$ would minimize $\alpha_k$ in (1), which is equivalent to:

$$\hat{v}_{po}^* = \arg\max_{\hat{v}_{po} \in C_k} \{(\hat{v}_{po} \cdot u_k)^2 - (\hat{v}_{po} \cdot \hat{v}_{po}) \cos^2(\alpha_k)\}. \quad (6)$$

If the geometric assumption (2) holds, a greedy clustering algorithm (Alg. 1) returns correct clustering of partially observed data under certain conditions, and error is bounded as

$$\frac{\|\mathbf{V} - \mathbf{W}_{po}^* \mathbf{H}_{po}^*\|_F}{\|\mathbf{V}\|_F} \leq \max_{k \in [K]} \{\sin 2\alpha_k\}, \quad (7)$$

where $\mathbf{W}_{po}^*$ and $\mathbf{H}_{po}^*$ are found by Alg. 2. Experiments show that this algorithm performs well on biological and imaging data in comparison to more complex methods (Chen & Varshney, 2019).

---

**Algorithm 1** Greedy Clustering with Missing Values

---

**Input** $\mathbf{V} \in \mathbb{R}_+^{F \times N}$, $K \in \mathbb{N}$, $\Omega \in \{0,1\}^{F \times N}$

**Output** $J \in \{0,1,...,K\}^N$; $\alpha \in (0, \pi/2)^K$; $u \in \mathbb{R}_+^{F \times K}$

1: Partition columns in $\mathbf{V}$ into subsets $\mathbf{V}_{fo}$ and $\mathbf{V}_{po}$, where $\mathbf{V}_{fo}$ contains data columns for which $\sum_i r_{ij} = F$, and $\mathbf{V}_{po}$ contains remaining columns.
2: Normalize $\mathbf{V}_{fo}$ so that all columns have unit $\ell_2$-norm. Let $\mathbf{V}'_{fo}$ be the normalized matrix
3: Cluster items in $\mathbf{V}'_{fo}$ using greedy clustering (See Alg. 1 from (Liu & Tan, 2017)) to obtain cluster indices $J$ and obtain minimax centers $u_1, ..., u_k$ from $W^*$.
4: **for** $v_{po} \in \mathbf{V}_{po}$ **do**
5:    Let $\Omega_j$ correspond to observed entries of $\mathbf{v}_{po}$. Find $k = \arg\max_{j \in [K]} \cos^{-1}\left(\frac{P_\Omega(\mathbf{z}_j) \cdot P_\Omega(\mathbf{v})}{\|P_\Omega(\mathbf{z}_j)\| \|P_\Omega(\mathbf{v})\|}\right)$. If this condition is maximized by more than one $k$, choose one at random. Add the index of $v_{po}$ to $J_k$.
6: **end for**
7: **for** $k \in [K]$ **do**
8:    $\alpha_k = \max_{v_{po}} \cos^{-1}\left(\frac{P_\Omega(v_{po}) \cdot P_\Omega(u_k)}{\|P_\Omega(v_{po})\| \|P_\Omega(u_k)\|}\right)$
9: **end for**
10: Return cone indices $J$, $u$, $\alpha$

---

**Algorithm 2** Rank-1 NMF with Missing Values

---

**Input** $\mathbf{V} \in \mathbb{R}_+^{F \times N}$, $\Omega \in \{0,1\}^{F \times N}$, $K \in \mathbb{N}$

**Output** $\hat{\mathbf{W}}^* \in \mathbb{R}_+^{F \times K}$ and $\hat{\mathbf{H}}^* \in \mathbb{R}_+^{K \times N}$

1: Cluster data using Alg. 1
2: Impute data using (4)
3: Perform rank-1 NMF on imputed data using Alg. 2 from (Liu & Tan, 2017)

---

## 3. Probabilistic error

We now make some probabilistic assumptions on our data and missingness patterns to calculate the expected maximum error of optimal recovery imputation. First, consider a cone $C$ in an $F$-dimensional space defined by $u$ and $\alpha$. Let us ignore the length of the vectors in $C$ and preserve only the angles of the vectors from $u$. We can then represent vectors of an $F$-dimensional cone as points in an $(F-1)$-dimensional ball. For example, a 3-dimensional cone can be represented as points in a circle.

Let there be $N$ points $\{x_1, ..., x_N\} \in \mathbb{R}^F$, drawn uniformly at random from $K$ $F$-dimensional balls, labeled $B_1, ..., B_K$. Let $d(x_i, x_j)$ be the Euclidean distance between $x_i$ and $x_j$. We assume there is at least one data point in each ball, and that

$$\max_{i,j \in B_k} d(x_i, x_j) < \min_{i \in B_k, j \notin B_k} d(x_i, x_j), \text{ for all } k = 1, ..., K.$$

This is equivalent to the geometric assumption in (2), and we can correctly cluster any points drawn from such balls

using Alg. 1. After obtaining the clusters, we compute the minimum covering sphere (MCS) on the points in each cluster (Hopp & Reeve, 1996). This gives us $K$ balls with $N_k$ points in each ball.

Now suppose we have partially observed entries in our data. Let the missingness of a point be a Bernoulli($\gamma$) random variable. That is, $x$ is fully observed with probability $\gamma$ and partially observed with probability $1 - \gamma$. There is now some uncertainty about the position of partially observed data points, so we find the MCS for only the fully observed points. This is analogous to step 3 in Alg. 2. By calculating the expected change in the radius of the MCS, we can calculate the expected change in its corresponding cone.

**Theorem 1** (Probabilistic bound on NMF error). *Given the setting described above, and assuming that the $N$ points are drawn uniformly at random from the $K$ balls. Suppose the points are randomly distributed along the radius of the $F$-ball and we pick points to be partially observed uniformly at random. Then after imputing with Alg. 1, we can tighten the bound in (7) to*

$$\frac{\|\mathbf{V} - \mathbf{W}_{po}^* \mathbf{H}_{po}^*\|_F}{\|\mathbf{V}\|_F} \le \max_{k \in [K]} \{\sin \alpha_k\}. \quad (8)$$

*Proof.* If the $N$ points are drawn uniformly at random from the $K$ balls, then $\mathbb{E}[N_k] = N/k$, and the expected number of fully observed and partially observed points in each cluster is $\mathbb{E}[|X_{k,fo}|] = \gamma N_k$ and $\mathbb{E}[|X_{k,po}|] = (1 - \gamma)N_k$.

Clearly, the volume of the MCS can only decrease as $|X_{k,fo}|$ decreases. Let $R_{max}$ be the radius of MCS if there were no missing values, and let $\hat{R}$ be the radius of the MCS of only the fully observed points. Then $\hat{R} < R_{max}$ only if any $x \in X_{po}$ originally lay on the surface of $\text{MCS}_{k,fo}$.

Let

$$N_{po} = \lceil (1 - \gamma)N \rceil. \quad (9)$$

Assume $x_i$ are i.i.d. and uniformly distributed (without loss of generality) on $[0, 1]$. This matches the assumption in the probabilistic analysis in (Liu & Tan, 2017) that the angles are drawn uniformly at random on $[0, \alpha]$ (see Fig. 2). Assuming a continuous distribution, almost surely no two points have exactly the same radius, and the probability of picking the $\ell$ outermost points is

$$\mathbb{P}(\ell) = \binom{N - \ell}{N_{po} - \ell} \bigg/ \binom{N}{N_{po}}, \text{ where } \ell = 0, 1, ..., N_{po}. \quad (10)$$

This gives us

$$\mathbb{E}[\ell] = \sum_{\ell=1}^{N_{po}} \ell \cdot \mathbb{P}[\ell] = \frac{1}{\binom{N}{N_{po}}} \sum_{\ell=1}^{N_{po}} \ell \cdot \binom{N - \ell}{N_{po} - \ell} \quad (11)$$

$$= \frac{\binom{N-1}{N_{po}-1} N(N+1)}{\binom{N}{N_{po}}(N - N_{po} + 1)(N - N_{po} + 2)}, \quad (12)$$

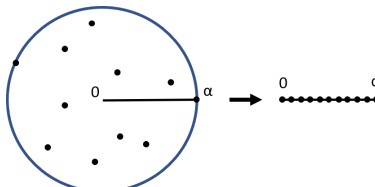

*Figure 2.* Assumption that points are uniformly random on the radius.

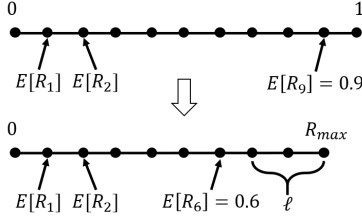

*Figure 3.* Example of $\mathbb{E}[\hat{R}]$ with $N = 9$ and $\ell = 3$.

where $N_{po}$ is dependent on $\gamma$, as defined in (9).

The radius of the resulting MCS is dependent on the distribution of points along the radius. We can determine $\hat{R}$ using order statistics. If we assume uniform distribution between 0 and 1, and order the points $x_1, \ldots, x_n$ so that $x_1$ is closest to the center of the sphere and $x_n$ is farthest, the radius of the $n$th point, $R_n$, is given by the beta distribution $R_n \sim B(n, 1)$, and $\mathbb{E}[R_n] = \frac{n}{n+1}$.

Thus if $\ell$ of the outermost points are chosen to be missing,

$$\mathbb{E}[\hat{R}] = R_{max} - (\ell/N)R_{max} = \left(\frac{N - \ell}{N}\right)R_{max} . \quad (13)$$

We illustrate with an example in Fig. 3. We can substitute $\mathbb{E}[\ell]$ for $\ell$, and since $\mathbb{E}[\ell]$ is a function of $\gamma$, we have derived the expected radius of the MCS as a function of missingness:

$$\mathbb{E}[\hat{R}] = \left(\frac{N - \mathbb{E}[\ell]}{N}\right)R_{max} . \quad (14)$$

Now we reverse the arrow in Fig. 2. Due to the random distribution of points in the sphere, removing the $\ell$ outermost points does not change the expected center $u$ of the MCS. Transitioning from spheres back to cones, we get

$$\mathbb{E}[\hat{\alpha}] = \left(\frac{N - \mathbb{E}[\ell]}{N}\right)\alpha . \quad (15)$$

Thus $\alpha - \mathbb{E}[\hat{\alpha}] = \frac{\mathbb{E}[\ell]}{N} \cdot \alpha$, and the normalized Frobenius distance between $\mathbf{W}_{fo}^*\mathbf{H}_{fo}^*$ and $\mathbf{W}^*\mathbf{H}^*$ for a single cone is:

$$\frac{\|\mathbf{W}_{fo}^*\mathbf{H}_{fo}^* - \mathbf{W}^*\mathbf{H}^*\|_F}{\|\mathbf{W}^*\mathbf{H}^*\|_F} \leq \sin\left(\frac{\mathbb{E}[\ell]}{N} \cdot \alpha\right). \quad (16)$$

If we assume $v_n \in \mathbf{V}$ are MCAR, the statistical mean of $\mathbf{V}_{fo}$ is the same as that of $\mathbf{V}$. Since $v_n$ are uniformly distributed, the range of $v_n$ remains centered on the mean, so the expected center of the MCS does not change. Thus the maximum difference between a point $v \in C_k$ and its imputed point $\hat{v}$ is $\sin\alpha_k$, and the theorem follows. $\square$

**Theorem 2** (Probabilistic bound on NMF error with different assumption). *If instead we assume points are uniformly distributed in the volume of the ball, we can find the corresponding change in radius. The expected NMF error is*

$$\mathbb{E}\left[\frac{\|\mathbf{V} - \mathbf{W}^*\mathbf{H}^*\|_F}{\|\mathbf{W}^*\mathbf{H}^*\|_F}\right] = \sin\left(\mathbb{E}[\hat{R}] \cdot \alpha\right). \quad (17)$$

*Proof.* See Appendix A in supplemental materials. $\square$

**Theorem 3** (Probabilistic bound on NMF error with normalized data). *If the data is normalized such that each vector has an $L_2$ norm of 1,*

$$\mathbb{E}\left[\frac{\|\mathbf{V} - \mathbf{W}^*\mathbf{H}^*\|_F}{\|\mathbf{W}^*\mathbf{H}^*\|_F}\right] = \sin\left(\mathbb{E}[\alpha^{po}]\right). \quad (18)$$

*Proof.* See Appendix B in supplemental materials. $\square$

## 4. Discussion

We gave a probabilistic error analysis of a clustering algorithm after minimax imputation. This analysis style can be extended to other clustering and imputation algorithms; various applications may require different model assumptions.

We now discuss the minimax approach and its implications on fairness. Missingness patterns themselves may carry information (Ghorbani & Zou, 2018), and statistics-based imputation methods may introduce unfairness (Martinez-Plumed et al., 2019). In certain social contexts, biases in algorithms can lead to unfair policy-making (Williams et al., 2018). Researchers attempt to mitigate some of these biases using multiple imputation (Azur et al., 2011) or weighted estimators (Chen et al., 2019). Philosopher John Rawls argues that in an effort to provide all individuals with equal opportunities, inequalities should only exist if they result in the worst off being better off (Rawls, 1971). In a scenario where one's place in society is chosen at random (including social status and other assets), one would prefer to land in a society that plays by a minimax rule, where the disadvantage of the worst off is minimized.

Future work aims to study how minimax imputation impacts fairness in decision-making and clustering (Chierichetti et al., 2017).

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

# Supplemental Materials
# Optimal recovery of missing values for non-negative matrix factorization: A probabilistic error bound

Rebecca Chen   Lav R. Varshney

## Appendix A

### Theorem: Probabilistic bound on NMF error with different assumption

If instead we assume points are uniformly distributed in the volume of the ball, we find the change in radius as follows. the expected NMF error is

$$\mathbb{E}\left[\frac{\|\mathbf{V} - \mathbf{W}^*\mathbf{H}^*\|_F}{\|\mathbf{W}^*\mathbf{H}^*\|_F}\right] = \sin\left(\mathbb{E}[\hat{R}] \cdot \alpha\right). \qquad (1)$$

*Proof.* If points are uniformly distributed in the volume of the ball, we find the change in radius as follows. First, calculate the volume of a $F$-dimensional ball of radius $R = 1$:

$$V_F(R) = \frac{\pi^{F/2}}{\Gamma(F/2 + 1)} R^F. \qquad (2)$$

Then we calculate radius $\hat{R}$ of an $F$-dimensional ball as:

$$\hat{R}_F(\hat{V}) = \frac{\Gamma(F/2 + 1)^{1/F}}{\sqrt{\pi}} \hat{V}^{1/F}, \qquad (3)$$

where volume $\hat{V} = \left(\frac{1-\ell}{N}\right) V_F(1)$.

The probability that a point $x$ is in $\mathrm{MCS}_{po}$ is $\frac{V(\hat{R})}{V(R_{max})}$.

Thus the expected radius given a missing parameter $\gamma$ is given by

$$\mathbb{E}[\hat{R}] = \hat{R}_F\left(\frac{1 - \mathbb{E}[\ell]}{N} V_F(1)\right), \qquad (4)$$

where $\mathbb{E}[\ell]$ is a function of $\gamma$, and the expected NMF error is

$$\mathbb{E}\left[\frac{\|\mathbf{V} - \mathbf{W}^*\mathbf{H}^*\|_F}{\|\mathbf{W}^*\mathbf{H}^*\|_F}\right] = \sin\left(\mathbb{E}[\hat{R}] \cdot \alpha\right). \qquad (5)$$

$\square$

## Appendix B

### Theorem: Probabilistic bound on NMF error with normalized data

If the data is normalized such that each vector has an $L_2$ norm of 1,

$$\mathbb{E}\left[\frac{\|\mathbf{V} - \mathbf{W}^*\mathbf{H}^*\|_F}{\|\mathbf{W}^*\mathbf{H}^*\|_F}\right] = \sin\left(\mathbb{E}[\alpha^{po}]\right). \qquad (6)$$

*Proof.* If the data is normalized such that each vector has an $L_2$ norm of 1, all the points will fall on the surface of a sphere. Let there be $N$ points $\{x_1, \ldots, x_N\} \in \mathbb{R}^F$, drawn at random from $K$ $F$-dimensional spherical caps of a radius $R$ $F$-ball, labeled $C_1, \ldots, C_K$. Let $d(x_i, x_j)$ be some distance between $x_i$ and $x_j$. Assume there is at least one data point in each spherical cap, and that our geometric assumption holds.

The area of an $F$-dimensional spherical cap is

$$A(R, h) = \frac{1}{2} A_F R^{F-1} I_{2rh-h^2/r^2}\left(\frac{F-1}{2}, \frac{1}{2}\right), \quad (7)$$

where $0 \le h \le R$, $A_n = 2\pi^{n/2}/\Gamma[n/2]$ is the area of the unit n-ball, $h$ is the height of the cap, which can be calculated as a function of the angle $\alpha$ between the center and the edge of the cap, and $I_x(a, b)$ is the regularized incomplete beta function. Using the same style of analysis from the previous section, we can find the expected angle $\mathbb{E}[\alpha^{po}]$ given a parameter $\gamma$ for partially observed points. Thus,

$$\mathbb{E}\left[\frac{\|\mathbf{V} - \mathbf{W}^*\mathbf{H}^*\|_F}{\|\mathbf{W}^*\mathbf{H}^*\|_F}\right] = \sin\left(\mathbb{E}[\alpha^{po}]\right). \qquad (8)$$

$\square$