# OpenReview forum: "Optimal recovery of missing values for non-negative matrix factorization: A probabilistic error bound"
_ICML.cc/2020/Workshop/Artemiss — ICML Artemiss 2020_

### Official Review · AnonReviewer1 · 2020-06-18
**Good paper; could be made clearer**

**Rating:** 7
**Confidence:** 4

**Review:**

Overall, I found this paper enjoyable to read although there are some exposition issues, which I encourage the authors to address.

Some exposition issues:
- Right after equation (1), perhaps it would be helpful to clearly define the notation you're using for the cluster centers and "radii"/angles rather than have the reader infer that this is what $u_k$ and $\alpha_k$ refer to (along with $C_k$).
- Equation (3) is stated in a perhaps peculiar way. From the text, I think you mean to take the union across $k$? Making this clear in the notation for both equations (3) and (4) would be helpful.
- In Section 3, at the end of the first paragraph, you say that vectors in an $F$-dimensional cone can be represented as $(F-1)$-dimensional balls, and then in the following paragraph, you say that you assume points to be drawn from $F$-dimensional balls. Is this a typo, i.e., are the points supposed to be drawn from $(F-1)$-dimensional balls?

Separately, I think it would be extremely helpful if the authors could much more clearly state what is new vs what is already in their earlier 2019 paper. From glancing at the earlier paper, there is substantial material overlap, and this newer paper constantly references the older paper. From my understanding, what's new is the tightening of a bound to drop a constant factor of 2, and separately also adding Theorems 2 and 3. Better motivating these additional theoretical guarantees would be helpful, along with numerical experiments to verify the theory. For example, in practice, how tight is bound (8)? Equations (17) and (18) are for expectations; how much variability is there about these expectations in practice? How are these related to real data?

Lastly, while the paper says that no assumption is made on the missingness (MCAR, MAR, MNAR, etc), I think that the missingness pattern is important in practice especially in a context where the vast majority of entries are missing (e.g., recommendation systems). Especially in the MNAR setting, I think it's possible coming up with examples where which entries are observed are biased in such a way that we get tricked and using local methods to impute could lead to incorrect imputations/downstream matrix factorization results; of course, this would correspond to the separability assumption that you use being violated. Numerical experiments to carefully assess how the proposed approach works under these different missing data mechanisms with varying amounts of missingness would strengthen the paper.

---

### Official Review · AnonReviewer2 · 2020-06-24
**Optimal recovery of missing values for non-negative matrix factorization: A probabilistic error bound**

**Confidence:** 5
**Rating:** 7

**Review:**

The paper focuses on the evaluation of the performance of Non negative Matrix Factorization (NMF) after imputation. In particular, the authors propose an imputation algorithm combining clustering and optimal recovery ; then, NMF is performed on the imputed data. The main contribution of the paper is to characterize the error of NMF, and to compare it quantatively to the error resulting from the same algorithm applied on fully observed data. In addition, the authors provide such characterization for different sets of assumptions.

---

### Decision · Program_Chairs · 2020-07-02

**Decision:**

Accept

**Comment:**

We're happy to accept this paper at Artemiss. We'll contact you soon to inform you about more details concerning the format of your presentation at the workshop, and the camera-ready version deadline. Please take into account the referee's comments to write the camera-ready version.